# Healthcare providers' perceived support from their organization is associated with lower burnout and anxiety amid the COVID-19 pandemic

**Katherine M. Reitz** [1], **Lauren Terhorst**[2,3], **Clair N. Smith** [4], **Insiyah K. Campwala**[5], **Maryanna S. Owoc** [5], **Stephanie M. Downs-Canner**[6], **Emilia J. Diego**[1], **Galen E. Switzer**[3,7,8,9], **Matthew R. Rosengart**[1,3], **Sara P. Myers** [1]*

1 Department of Surgery, University of Pittsburgh, Pittsburgh, PA, United States of America, 2 School of Health and Rehabilitation Sciences, University of Pittsburgh, PA, United States of America, 3 Department of Clinical and Translational Science, University of Pittsburgh, PA, United States of America, 4 Department of Orthopaedic Surgery, University of Pittsburgh, Pittsburgh, PA, United States of America, 5 School of Medicine, University of Pittsburgh, Pittsburgh, PA, United States of America, 6 Department of Surgery, University of North Carolina-Chapel Hill School of Medicine, Chapel Hill, NC, United States of America, 7 Department of General Medicine, University of Pittsburgh, Pittsburgh, PA, United States of America, 8 Department of Psychiatry, University of Pittsburgh, Pittsburgh, PA, United States of America, 9 Center for Health Equity Research and Promotion, Veterans Affairs Pittsburgh Healthcare System, Pittsburgh, PA, United States of America

* myerssp@upmc.edu

**Data Availability Statement:** All relevant data are within the paper and its Supporting Information files.

## Abstract

### Background

Professional burnout represents a significant threat to the American healthcare system. Organizational and individual factors may increase healthcare providers' susceptibility or resistance to burnout. We hypothesized that during the COVID-19 pandemic, 1) higher levels of perceived organizational support (POS) are associated with lower risk for burnout and anxiety, and 2) anxiety mediates the association between POS and burnout.

### Methods

In this longitudinal prospective study, we surveyed healthcare providers employed full-time at a large, multihospital healthcare system monthly over 6 months (April to November 2020). Participants were randomized using a 1:1 allocation stratified by provider type, gender, and academic hospital status to receive one of two versions of the survey instrument formulated with different ordering of the measures to minimize response bias due to context effects. The exposure of interest was POS measured using the validated 8-item Survey of POS (SPOS) scale. Primary outcomes of interest were anxiety and risk for burnout as measured by the validated 10-item Burnout scale from the Professional Quality (Pro-QOL) instrument and 4-item Emotional Distress-Anxiety short form of the Patient Reported Outcome Measurement Information System (PROMIS) scale, respectively. Linear mixed models evaluated the associations between POS and both burnout and anxiety. A mediation analysis evaluated whether anxiety mediated the POS-burnout association.

**Funding:** Dr. Katherine Reitz was supported in part by the National Heart, Lung, and Blood Institute (5T32HL0098036) and National Institutes of Health (L30AG064730).

**Competing interests:** No author have competing interests.

## Results

Of the 538 participants recruited, 402 (75%) were included in the primary analysis. 55% of participants were physicians, 73% 25–44 years of age, 73% female, 83% White, and 44% had ≥1 dependent. Higher POS was significantly associated with a lower risk for burnout (-0.23; 95% CI -0.26, -0.21; p<0.001) and lower degree of anxiety (-0.07; 95% CI -0.09, -0.06; p = 0.010). Anxiety mediated the associated between POS and burnout (direct effect -0.17; 95% CI -0.21, -0.13; p<0.001; total effect -0.23; 95% CI -0.28, -0.19; p<0.001).

## Conclusion

During a health crisis, increasing the organizational support perceived by healthcare employees may reduce the risk for burnout through a reduction in anxiety. Improving the relationship between healthcare organizations and the individuals they employ may reduce detrimental effects of psychological distress among healthcare providers and ultimately improve patient care.

## Introduction

Frontline healthcare providers have been disproportionately affected by coronavirus disease 2019 (COVID-19) in their places of employment [1]. The global health crisis incited by COVID-19 has had detrimental consequences for mental health and, specifically, may potentiate provider burnout [1]. Professional burnout, characterized by emotional exhaustion, career de-prioritization, and loss of self-efficacy, represents a significant threat to the American healthcare system [1, 2].

Burnout among healthcare providers is a prevalent and well-documented phenomenon [3] and has been associated with adverse clinical outcomes [4], reduced productivity [5], and increased rate of medical errors [6]. Aside from the consequences to providers and their patients, burnout represents a serious financial burden; $4.6 billion lost annually is attributed to physician attrition and loss of clinical hours secondary to burnout [6]. Heavy workloads, changing clinical roles, reduced decision latitude (i.e., the ability to exercise control over work-related responsibilities), and lack of support from supervisors have been cited as causes of increased occupational stress [7], a key predictor of burnout [8]. During this pandemic additional stressors such as lack of personal protective equipment and fear of contagion have been shown to augment the risk for burnout [1, 3]. These issues have threatened the stability of our healthcare organizations and, therefore, the COVID-19 medical crisis has also functionally become an *organizational crisis* [9]. As a result, health-related performance and quality outcomes may, in part, depend on how a healthcare organization manages to allocate resources and human capital [10] and whether it effectively develops policies to protect and support its employees.

While burnout has been described as a reaction to organizational factors such as work-related stress [11], individual factors such as sociodemographic factors that may be associated with development of coping skills in response to stressful situations (e.g., age), availability of support systems (marital status), and type and degree of responsibility (e.g., occupation, parental/caretaker status, etc.) may increase susceptibility to burnout [12] and/or anxiety [13]. It is, however, unknown whether the degree to which an individual feels supported by his/her/their employer, i.e., perceived organizational support (POS), a key driver of job satisfaction and

performance [6], influences anxiety and burnout under circumstances of crisis. In this longitudinal prospective study of healthcare providers, we aim to evaluate whether 1) POS is associated with anxiety or burnout, hypothesizing that during the COVID-19 pandemic higher levels of POS are associated with lower burnout and anxiety, and 2) anxiety mediates the association between POS and burnout. Understanding the relationship between these constructs is paramount to designing interventions that preserve the psychological wellbeing of healthcare providers, a requisite for maintaining a healthy and productive workforce [14], and delivering optimal patient care [15].

## Methods

### Participants, setting, study design

A longitudinal prospective survey study of healthcare providers from 20 community and academic hospitals within a single healthcare system in Pennsylvania was conducted to determine the association of POS with burnout and anxiety. In order to assess how pandemic-related changes in organizational dynamics and protocols may influence anxiety and risk for burnout over time, an online survey of previously validated measures was administered monthly for a six-month period (April 28 to October 11, 2020). This study was approved by the University of Pittsburgh Institutional Review Board (STUDY20040051).

Based on previous literature investigating burnout among healthcare providers during COVID-19, assuming 35% of providers experience burnout [1, 16], a minimum of 88 respondents were required to evaluate the association between POS and burnout, allowing for a 0.1 acceptable margin of error and alpha of 0.05 [1]. All full-time providers employed by the healthcare system were recruited for study participation via e-mail (S1 Appendix). E-mail addresses were abstracted by three study investigators (IS, MO, and SPM) from an internal employee electronic directory (April 1–14, 2020) which is available to all UPMC staff and is maintained by the institution's Office of Human Resources. In order to better understand how healthcare workers with differing levels of patient-care responsibilities may have reacted to this organizational and global health crisis, we sampled a variety of providers; attending physicians, physicians in training (i.e., residents, fellows), advanced practice providers (i.e., nurse practitioners, physician assistants), nurses, and other providers (i.e., respiratory therapists, patient care technicians) employed by an UPMC-associated hospital in Pennsylvania with sufficient exposure to work-place and institutional culture (i.e., >12 months of employment) were eligible for inclusion. As this study aims to assess reactions associated with increased risk of patient-care related COVID-19 exposure, healthcare workers whose primary patient contact was via telehealth were excluded. Participants were asked to verify aforementioned inclusion/exclusion criteria, consent to participation, and provide baseline demographic data via electronic enrollment form (S2 and S3 Appendices). Participants were provided with study details, purpose of the investigation, and incentive information in the text of the recruitment e-mail and within the consent document. Participants were made aware that although the investigators did not anticipate that the study would cause adverse events, responding to survey items may evoke emotional responses such as stress. There were no direct benefits to the participant for completing the surveys. Participants were free to withdraw from the study at any point; in this circumstance, all contact and data collection would cease and any existing data would be deleted upon request. Based on data obtained from this form, participants were then randomized using a 1:1 allocation stratified by provider type, gender, and academic hospital status to receive one of two versions of the survey instrument formulated with different ordering of the measures to minimize response bias due to context effects [17].Participants received the same version at all six time points via an automated email providing a link to a University of

Pittsburgh RedCap database. Alphanumeric codes were used to anonymize survey data for the purposes of blinding individuals conducting the study and statisticians. De-identified data from serial surveys were collected and stored for repeated measure. Each survey also assessed whether participants had experienced parameters that may be associated with COVID-19 and/ or its detrimental emotional effects including yes/no questions regarding perceived illness, illness necessitating time off from work, and strain on personal relationships. As we speculated that providers' degree of stress might correlate with COVID-19-related hospital admissions, we determined monthly per hospital caseload for each participant. Caseload was defined as the proportion of COVID-19 admissions per total hospital beds, abstracted from the healthcare system [18]. Survey participation to completion was incentivized by the random selection of two $250 and 20 $25 VISA® and Starbucks® gift cards.

## Study measures

POS was assessed using the validated, unidimensional 8-item Survey of POS (SPOS) scale [19, 20]. Items appraise the degree to which an individual agreed with statements describing support or commitment from their employer organization in the 30 days prior using a 7-point Likert scale (0 = Strongly disagree, 6 = Strongly agree; scale minimum = 0, maximum = 42). Risk for burnout was evaluated using the 10-item Burnout scale from the Professional Quality (Pro-QOL) scale. This instrument is a validated, multidimensional metric that assesses consequences of stress a provider may experience from exposure to patients in emotional or physical distress [21, 22]. Participants reflect on the frequency with which they experience hopelessness or difficulty completing job-related tasks effectively in the prior 30-days using a 5-point Likert scale (1 = Never, 5 = Very often; scale minimum = 5, maximum = 30). General anxiety was assessed using the 4-item Emotional Distress-Anxiety short form of the validated, multidimensional Patient Reported Outcome Measurement Information System (PROMIS) [23] scale that evaluates the frequency to which anxiety-related symptoms were experienced in the prior 7-days using a 5-point Likert scale (1 = Never, 5 = Very often; scale minimum = 7, maximum = 35). Each individual scale quantifies the psychosocial construct by the summation of all responses to Likert items. For each scale, higher total scores represent more of the psychosocial construct. Internal consistency of each scale or subscale are as follows: 8-item SPOS = 0.93 [24], Burnout = 0.84 [25], and Anxiety = 0.97 [26].

## Statistical analysis plan

Descriptive statistics were generated for demographics, summary scales, and additional variables of interest. As a means of managing missingness unlikely to be at random and due to loss to follow-up, we restricted our primary analysis to include survey time points with at least 75% participant retention [27]. Thus, participants who completed at least two of the three first surveys constituted the cohort included in the primary analysis. Demographic data and variables of interest were compared between participants receiving version A and B, and between cohorts included and excluded from the primary analysis.

The association of POS, burnout, and anxiety were examined using linear mixed models that included fixed effect of time and random effects for hospital and participant [28]. Multivariable models additionally adjusted for covariates that were significantly associated with burnout or anxiety on univariable analysis and COVID-19 caseload. Further exploratory analyses, agreed upon *a priori*, were performed to determine how demographic variables moderate associations between POS, anxiety, and burnout. Sensitivity analyses were performed to assess how associations in the primary analysis between POS and outcomes of interest may have differed when using data from all six survey administrations. We performed a mediation analysis

for each of the first three survey administrations with 5000 bootstrapped samples to evaluate if anxiety was the mechanism through which POS transmits its effect on burnout [29].

Analysis, completed with SAS 9.4 (SAS Institute Inc.; PROCESS macro) or PRISM 7.0 (GraphPad), and data presentation were compliant with the Strengthening the Reporting of Observational Studies in Epidemiology guidelines (S3 Appendix) [30]. Statistical significance was considered below a threshold p-value of 0.05.

## Results

538 individuals completed the enrollment form. 516 were eligible for study inclusion and participated in survey 1. Internal consistency calculated for subscales was as follows: SPOS = 0.91, Burnout = 0.85, and Anxiety = 0.89. The primary analysis included data from 402 participants who completed surveys 1–3 (Fig 1). Demographic data are presented in Table 1. Demographics did not differ based on survey version (S1 Table). Compared to participants whose data was excluded from the primary analysis, a greater proportion of participants included were married or living like married, White persons, and had income ≥$163,301 (S2 Table). Those included in the primary analysis reported more perceived illness and time off from work, but less relationship strain than participants excluded (S1 Fig). Study measures (i.e., POS, risk of burnout, and anxiety) did not differ overtime or between groups. Age, gender, occupation, income, having dependents, perceiving illness, experiencing illness necessitating time off of work, and endorsing relationship strain were associated with either burnout or anxiety on univariate analysis and included in the adjusted model (Table 2).

### Burnout

Higher POS was significantly associated with a lower risk for burnout (Table 3). Having dependents was associated with a lower risk of burn out. Perceiving illness, endorsing strain on personal relationships, and being nurse (as compared with respiratory therapists and

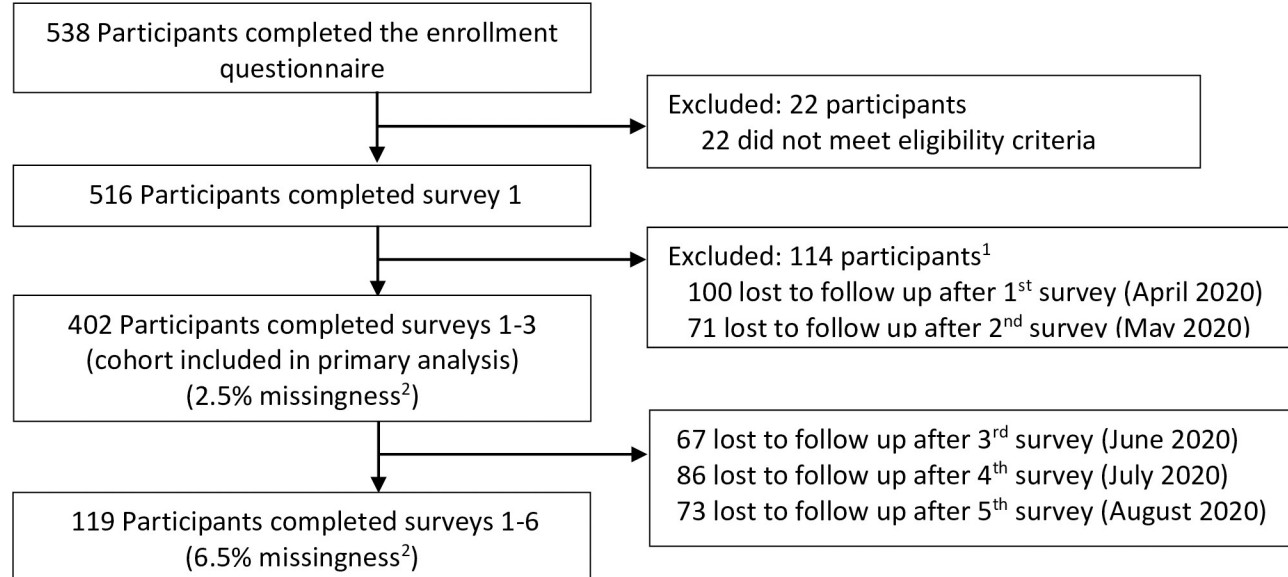

**Fig 1. Cohort accrual.** [1] Discrepancy between number excluded from primary analysis and number lost to follow-up stems from participants failing to complete consecutive surveys. As data from participants who completed two of the three initial surveys was included in the primary analysis, some individuals may have cursorily been considered lost to follow up after survey 1 but been included in the analysis if they returned to complete survey 3. [2] Within survey missingness.

**Table 1. Enrollment questionnaire for participants included in primary analysis.**

| Variable | | Primary Cohort (N = 402) |
|---|---|---|
| | | n (%) |
| Age, years[1] | | |
| <24 | | 31 (7.7) |
| 25–44 | | 292 (72.6) |
| 45–64 | | 71 (17.7) |
| 65+ | | 8 (2.0) |
| Male | | 110 (27.4) |
| Married or living like married | | 263 (65.4) |
| Race[1,2] | | |
| White American, European American, or Middle Eastern American | | 335 (83.3) |
| Black or African American | | 9 (2.2) |
| Asian American | | 39 (9.7) |
| Native Hawaiian or Other Pacific Islander | | 1 (0.3) |
| Other | | 8 (2.0) |
| Declined to answer | | 10 (2.5) |
| Hispanic ethnicity | | 9 (2.2) |
| Occupation | | |
| Attending physician | | 115 (28.6) |
| Trainee (resident/fellow) | | 139 (34.6) |
| Advanced practice provider | | 31 (7.7) |
| Nursing staff | | 62 (15.4) |
| Other staff[3] | | 55 (13.7) |
| Academic Hospital[4] | | 236 (58.7) |
| Income[1,2] | | |
| | $0–14,000 | 5 (1.2) |
| | $14,001–53,000 | 34 (8.5) |
| | $53,001–85,500 | 102 (25.4) |
| | $85,501–163,300 | 110 (27.4) |
| | $163,301–207,350 | 26 (6.5) |
| | $207351–518,400 | 94 (23.4) |
| | $518,041+ | 13 (3.2) |
| | Declined to answer | 18 (4.5) |
| Parent status[1,5] | | |
| No dependents | | 225 (56.0) |
| 1 dependent | | 66 (16.4) |
| 2 dependents | | 57 (14.2) |
| 3 dependents | | 35 (8.7) |
| 4+ dependents | | 18 (4.5) |
| Primary caretaker status[6] | | 125 (31.1) |

1 Categorization consistent with 2020 census.

2 Survey indicated categorical responses as optional, declining to answer was therefore not considered missing data.

3 Defined as respiratory therapist or patient care technician.

4 Defined as a tertiary care hospital that is organizationally integrates with a medical school and/or residency program.

5 Defined as having one or more child for whom the participant is a guardian.

6 Defined as serving as a primary caretaker for another individual.

**Table 2. Univariate associations between variables and outcomes of interest in the primary cohort (N = 402).**

| Variable | Burnout[1] | | | Anxiety[1] | | |
|---|---|---|---|---|---|---|
| | Coefficient | 95% CI | p-value | Coefficient | 95% CI | p-value |
| Age2, years | | | 0.070 | | | <0.001 |
| $\leq$24 | 1.87 | (0.28, 3.45) | | 2.28 | (1.41, 3.14) | |
| 25–44 | 0.55 | (-0.38, 1.48) | | 0.87 | 0.36, 1.38 | |
| 45+ | Ref | | | Ref | | |
| Male sex | -1.49 | (-2.33, -0.65) | 0.001 | -1.67 | (-2.13, -1.22) | <0.001 |
| Married/living like married | -0.50 | (-1.27, 0.27) | 0.200 | -0.31 | (-0.73, 0.12) | 0.150 |
| Race[2] | | | | | | |
| White, European, Middle-Eastern | 0.99 | (-0.02, 2.00) | 0.051 | 0.19 | (-0.37, 0.75) | 0.500 |
| Hispanic Ethnicity | -0.77 | (-3.26, 1.71) | 0.540 | 0.77 | (-0.58, 2.12) | 0.260 |
| Income[2] | | | 0.001 | | | <0.001 |
| $0–53,000 | 1.93 | (0.57, 3.28) | | 2.29 | (1.55, 3.02) | |
| $53,701–85,500 | 1.82 | (0.85, 2.79) | | 1.53 | (1.00, 2.06) | |
| $85,501–163,300 | 0.90 | (-0.06, 1.86) | | 0.72 | (0.19, 1.24) | |
| $163,301+ | Ref | Ref | | Ref | Ref | |
| Occupation | | | <0.001 | | | <0.001 |
| Attending | -3.06 | (-4.41, -1.72) | | -3.08 | (-3.75, -2.40) | |
| Resident/fellow | -1.85 | (-3.57, -0.13) | | -1.93 | (-2.61, -1.25) | |
| Advanced practice provider | -1.76 | (-3.44, -0.08) | | -1.51 | (-2.39, -0.64) | |
| Nursing staff | 0.51 | (-0.86, 1.88) | | -0.09 | (-0.82, 0.64) | |
| Other staff [3] | Ref | Ref | | Ref | Ref | |
| Parent status[2,4] (Reference, Not parent) | -1.20 | (-1.94. -0.46) | 0.002 | -0.97 | (-1.37, 0.56) | <0.001 |
| Primary caretaker[5] (Reference, No primary caretaker) | -0.05 | (-0.85, 0.75) | 0.900 | -0.01 | (-0.45, 0.43+ | 0.960 |
| Academic hospital[6] (Reference, Not academic hospital) | -0.39 | (-0.54, 0.60) | 0.600 | 0.33 | (-0.76, 1.42) | 0.550 |
| Perceived illness[1] (Reference, No perceived illness) | 4.93 | (4.00, 5.87) | <0.001 | 2.47 | (1.96, 2.99) | <0.001 |
| Time taken off for illness[1] (Reference, No time off) | 4.45 | (2.95, 5.95) | <0.001 | 2.48 | (1.66, 3.31) | <0.001 |
| Relationship strain[1] (Reference, No relationship strain) | 5.47 | (4.81, 6.14) | <0.001 | 2.67 | (2.29, 3.04) | <0.001 |
| COVID-19 Caseload[1,7] | -0.01 | (-0.20,0.19) | 0.186 | 0.03 | (-0.12, 0.18) | 0.176 |

1 Repeated measures evaluated at each survey timepoint. P-values correspond to the Wald Test for each correlation.

2Categorization consistent with 2020 census. Categories consolidated if reported by <5% of respondents.

3 Defined as respiratory therapist or patient care technician.

4 Defined as having one or more child for whom the participant is a guardian.

5 Defined as serving as a primary caretaker for another individual.

6 Defined as a tertiary care hospital that is organizationally integrates with a medical school and/or residency program.

7 Proportion of COVID-19 cases per maximal bed capacity at the identified hospital of employment by the participants.

Values are from linear mixed models with fixed effects accounting for time and the single variable of interest as well as random effects accounting for the hospital and participant.

Abbreviations: CI, confidence intervals.

patient care technicians as the reference category) were associated with a higher risk of burn out (Table 3). The association between higher POS and lower risk of burnout was consistently observed, both across all subgroups and in sensitivity analysis (S3 and S4 Tables). In addition

to the variables that were significantly associated with burnout in the primary analysis, male gender was significantly associated with reduced risk of burnout in the sensitivity analysis (S4 Table).

## Anxiety

Higher POS was associated with lower anxiety (Table 3). Reporting perceived illness and endorsing circumstantial strain on personal relationships were associated with higher anxiety. Attending and trainee physician status were associated with lower anxiety. Having dependents and male sex were also associated with lower anxiety. The association between POS and anxiety was consistently observed across subgroups, but the effect was significantly moderated by occupation, relationship strain, and income. The magnitude of the inverse relationship was

**Table 3. Adjusted analysis between variables and outcomes of interest in the primary cohort (N = 402).**

| | Burnout[1] | | | Anxiety[1] | | |
|---|---|---|---|---|---|---|
| | Coefficient | 95% CI | p-value | Coefficient | 95% CI | p-value |
| Perceived organizational [1] | -0.23 | (-0.26, -0.21) | < .0001 | -0.07 | (-0.09, -0.06) | <0.001 |
| Age2, years (Reference, 45+) | | | | | | |
| <25 | -0.42 | (-1.29, 0.46) | 0.350 | 0.20 | (-0.33, 0.73) | 0.460 |
| 25–44 | -1.28 | (-2.84, 0.28) | 0.109 | -0.58 | (-1.02, -0.14) | 0.009 |
| Male Sex | -0.73 | (-1.47, 0) | 0.050 | 0.94 | (0.50, 1.39) | <0.001 |
| Occupation (Reference, other staff[3]) | | | | | | |
| Attending | -1.18 | (-2.79, 0.44) | 0.152 | -2.08 | (-3.06, -1.10) | <0.001 |
| Trainee (resident/fellow) | -0.58 | (-1.98, 0.82) | 0.414 | -1.44 | (-2.28, -0.60) | 0.001 |
| Advanced practice provider | -1.25 | (-2.87, 0.37) | 0.129 | -0.94 | (-1.92, 0.05) | 0.063 |
| Nursing staff | -1.32 | (-2.59, -0.04) | 0.044 | -0.74 | (-1.51, 0.04) | 0.063 |
| Income (Reference, $163,301+) | | | | | | |
| $0–53,000 | -0.07 | (-1.67, 1.53) | 0.931 | -0.05 | (-1.03, 0.92) | 0.916 |
| $53,701–85,500 | 0.07 | (-1.11, 1.24) | 0.913 | 0.08 | (-0.63, 0.78) | 0.832 |
| $85,501–163,300 | 0.05 | (-1.07, 1.16) | 0.934 | -0.48 | (-1.15, 0.19) | 0.161 |
| Parent status[4] (Reference, Not parent) | -1.17 | (-1.9, -0.44) | 0.002 | -0.58 | (-1.02, -0.14) | 0.009 |
| Perceived illness[1](Reference, No perceived illness) | 2.55 | (1.58, 3.52) | < .0001 | 0.93 | (0.35, 1.51) | 0.002 |
| Time taken off for illness[1] (Reference, No time off) | 0.47 | (-0.95, 1.88) | 0.521 | 0.61 | (-0.24, 1.47) | 0.159 |
| Relationship strain[1](Reference, No relationship strain) | 3.87 | (3.25, 4.48) | < .0001 | 1.99 | (1.62, 2.36) | <0.001 |
| Survey timeframe (Reference, April) | | | | | | |
| May | -0.26 | (-0.98, 0.46) | 0.480 | -0.42 | (-0.9, 0.06) | 0.087 |
| June | 0.24 | (-0.47, 0.94) | 0.507 | -0.41 | (-0.85, 0.03) | 0.065 |
| COVID-19 Caseload[1,5] | -0.04 | (-0.46, 0.37) | 0.421 | 0.443 | (-0.10, 0.16) | 0.623 |
| Intercept | 31.99 | (30.02, 33.97) | < .0001 | -0.07 | (-0.09, -0.06) | <0.001 |

1 Repeated measures evaluated at each survey timepoint. P-values correspond to the Wald Test for each individual variable.

2 Categorization consistent with 2020 census. Categories consolidated if reported by <5% of respondents.

3 Defined as respiratory therapist or patient care technician.

4 Defined as having one or more child for whom the participant is a guardian.

5 Proportion of COVID-19 cases per maximal bed capacity at the identified hospital of employment by the participants.

Values are from linear mixed models with fixed effects accounting for time, POS, and all variables of interest that were significant in the univariate analysis, and with random effects accounting for the hospital and participant.

Abbreviations: CI, confidence intervals.

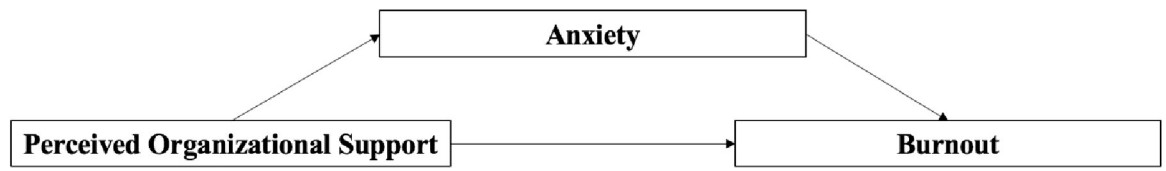

| Study Month[1] | Direct effect | | | Indirect effect | Total effect |
|---|---|---|---|---|---|
| | Perceived organizational support on anxiety | Anxiety on risk of burnout | Perceived organizational support on risk of burnout | Perceived organizational support on risk of burnout | Perceived organizational support on risk of burnout (Anxiety as mediator) |
| | *Effect size (95% CI)* | | | *Effect size (95% CI)* | *Effect size (95% CI)* |
| April | -0.17 (-0.21, -0.12) | 0.86 (0.71, 1.00) | -0.17 (-0.21, -0.13) | -0.06 (-0.09, -0.03) | -0.23 (-0.28, -0.19) |
| May | -0.16 (-0.20, -0.12) | 0.77 (0.62, 0.91) | -0.16 (-0.20, -0.12) | -0.05 (-0.08, -0.03) | -0.21 (-0.26, -0.17) |
| June | -0.18 (-0.22, -0.14) | 0.84 (0.68, 1.00) | -0.18 (-0.22, -0.14) | -0.07 (-0.10, -0.04) | -0.25 (-0.30, -0.20) |

**Fig 2. Anxiety mediates the association between perceived organizational support and burnout.** [1] Full adjusted models for each study month are available in S5–S7 Tables. Abbreviations: CI, confidence intervals.

strongest among respiratory therapists and patient care technicians compared with other providers, those with relationship strain compared to those without, and those in lower compared to the highest income bracket (S3 Table). Higher POS remained associated with lower anxiety in sensitivity analysis (S4 Table).

## Mediation analysis

Analysis of data from the first survey administration demonstrated that higher POS was associated with lower anxiety (Fig 2). Higher anxiety was associated with higher risk for burnout. Without adjusting for anxiety there was a significant direct effect of POS on burnout, such that higher POS was associated with lower risk for burnout. There was a negative indirect effect of POS on burnout through anxiety. The total of indirect and direct effects in the mediation analysis indicated that anxiety is a partial mediator of the association between POS and burnout. Mediation analysis of data from second and third survey administrations were similar (Fig 2; S5–S7 Tables).

## Discussion

Burnout is characterized by emotional exhaustion, depersonalization, and loss of self-worth [31]. Up to 50% of doctors and nurses report symptoms of burnout [32–34] with health care organizations cited as a leading cause [35–40]. In this longitudinal prospective survey-based study of healthcare workers during the COVID-19 pandemic, higher POS was associated with lower anxiety and risk for burnout. Anxiety was found to partially mediate the association between POS and burnout, suggesting that in the context of crisis, strategies aimed at alleviating anxiety might reduce burnout.

In our study, we observed that higher POS reduced risk for burnout. We speculate that this inverse relationship may reflect consequences that result from how organizations behave in response to periods of crisis. The *threat-rigidity effect* postulates that organizations may

centralize power and reduce communication in periods of crisis [41], which may, in turn, lead to reduced POS if the employee feels that his/her/their employer is not concerned about the employee's well-being [42]. By the same token, as our data would indicate, if employees perceive support from their organizations, risk of burnout may decrease.

While previous studies have demonstrated that providers frequently cite organizational causes of burnout [3, 36–40]; characteristics of the person may also contribute to burnout during periods of crisis [43]. We identified several individual factors that were associated with risk for burnout. Modifiable factors included strain on personal relationships. Our findings are consistent with data from others demonstrating that restrictions brought on by COVID-19 affect relationship quality and have consequences for mental health [44]. Other individual factors such as occupation and household income may also influence burnout [3]. Our finding that relative to respiratory therapists and patient care technicians, being a staff nurse is associated with increased risk of burnout reveals that provider-type may contribute to differences in pandemic-related burnout. We speculate that the degree to which individual factors influence burnout may vary depending on social support. The protective effect of having children that we and others have observed [45] and the aforementioned detrimental effect of COVID-19 related circumstances on personal relationships substantiate this theory. With respect to non-modifiable risk factors, the influence of gender on burnout in general remains unclear. In our study, male gender just reached the threshold of significance on primary analysis and was significantly associated with reduced burnout risk in the sensitivity analysis. Some studies have observed that the prevalence of burnout is higher in women [46], while others have shown that women have decreased risk of burnout [47]. These conflicting findings may indicate that confounders obfuscate whether gender is independently associated with burnout. One such issue, which is pertinent to our analysis, is the disparate gender composition among different provider-types. Although studies are inconsistent in their findings [48], lack of control over workplace practices, which is necessarily influenced by gender-based structural bias, may contributed to differences in anxiety and risk for burnout in specialties where women are underrepresented [48]. Additionally, it may be that women are at higher risk for certain facets, such as emotional exhaustion, whereas men are predisposed to others, such as depersonalization [49]. Furthermore, there is evidence that gender disparities in mental health may be exacerbated by the pandemic and that disproportionate distribution of domestic responsibilities may contribute [50]. Our observation that individual factors contribute to burnout risk echoes others' work showing that effective interventions may be tailored based on high-risk characteristics, personalized based on occupation, and/or focus on strengthening social support systems [51].

Although various psychological conditions have been found to contribute to burnout, the effect of anxiety, especially in acute periods of threat, is relatively understudied [52]. Anxiety has both stable (i.e., trait) and dynamic (i.e., state) components. Trait anxiety describes the degree to which an individual may be prone to anxiety [53]. State anxiety, on the other hand, is a reaction to circumstances that are perceived as threatening [53] such as the COVID-19 pandemic [54]. Our finding that anxiety partially mediates the effect of POS on burnout has been alluded to by others, who have theorized that work situations may trigger heightened anxiety and lead to burnout [55]. Additionally, existing literature is consistent with our observation that individual factors such as gender, occupation, and income may influence response to work-related stress and increase vulnerability to anxiety [56]. Understanding these relationships is the first step in developing interventions that address the multifaceted nature of burnout.

There are several strengths and limitations worth addressing. Although several provider-types were recruited in an attempt to understand how individuals with different patient-care

responsibilities may have been variably influenced by pandemic-related restrictions and we surveyed individuals from multiple hospitals, our data does represent a single healthcare system and geographical area, which may limit external validity. Efforts were made to reduce response bias by randomizing individuals to receiving one of two versions of our survey, which differed in the order of the constructs assessed. While conducting a prospective longitudinal study allowed for the ability to assess providers' responses to changes in organizational support, anxiety, and risk for burnout over time, this methodology also likely contributed to high attrition rates at later survey administrations, presumably due to survey fatigue. Therefore, missingness may not be at random, forcing respondent exclusion from the primary analysis. Further, the racial/ethnic homogeneity of our sample likely compromised our ability to accurately assess how sociodemographic factors influence burnout and anxiety in the context of a multifaceted construct such as POS. As this study was executed during the early months of the pandemic, variability in diagnostic testing across the healthcare system may have contributed to underreporting of hospital-specific rates of COVID-19. Finally, survey response was adequately powered to evaluate the *a priori* hypotheses; however, exploratory moderation analyses were underpowered.

## Conclusion

During a health crisis, increasing the organizational support perceived by healthcare employees may reduce the risk for burnout through a reduction in anxiety. Improving the relationship between healthcare organizations and the individuals they employ may reduce detrimental effects of psychological distress among healthcare providers and ultimately improve patient care. These data indicate that interventions aimed at increasing perceived organizational support, such as augmenting discretionary services and benefits (e.g., childcare, psychological-support services, etc.), or fostering transparency in the adoption and oversight of institutional policies in response to crisis, may promote provider well-being.

## Supporting information

**S1 Appendix. Recruitment E-mail.**
(DOCX)

**S2 Appendix. Consent form.**
(DOCX)

**S3 Appendix. Enrollment (baseline) survey.**
(DOCX)

**S4 Appendix. Strengthening the Reporting of Observational Studies in Epidemiology (STROBE).**
(DOCX)

**S1 Table. Demographics of participants by randomization group (i.e., survey version A vs. survey version B).**
(DOCX)

**S2 Table. Demographics of participants included and excluded from primary analysis.**
(DOCX)

**S3 Table. Subgroup analyses investigating whether association between perceived organizational support and outcomes of interest (i.e., burnout, anxiety) differ over categories of**

**covariates.**
(DOCX)

**S4 Table. Univariate associations between variables and outcomes of interest, sensitivity analysis using all time points.**
(DOCX)

**S5 Table. Meditation analysis, 1st survey (April 2020).**
(DOCX)

**S6 Table. Meditation analysis, 2nd survey (May 2020).**
(DOCX)

**S7 Table. Meditation analysis, 3rd survey (June 2020).**
(DOCX)

**S1 Fig. Repeated measure data and COVID-19 caseload over time.**
(DOCX)

**S1 Dataset.**
(XLSX)

## Acknowledgments

We would like to acknowledge Brenda Diergaarde, PhD Associate Professor, Human Genetics for sharing her expertise in the creation and use of the RedCap database. We would also like to thank Drs. Robert Handzel, Kristina Nicholson, Ariel Shensa, and Kenneth Lee for their support throughout the early phases of project development.

## Author Contributions

**Conceptualization:** Katherine M. Reitz, Insiyah K. Campwala, Maryanna S. Owoc, Stephanie M. Downs-Canner, Galen E. Switzer, Matthew R. Rosengart, Sara P. Myers.

**Data curation:** Katherine M. Reitz, Lauren Terhorst, Stephanie M. Downs-Canner, Emilia J. Diego, Galen E. Switzer, Sara P. Myers.

**Formal analysis:** Katherine M. Reitz, Lauren Terhorst, Clair N. Smith, Maryanna S. Owoc, Matthew R. Rosengart, Sara P. Myers.

**Investigation:** Katherine M. Reitz, Insiyah K. Campwala, Stephanie M. Downs-Canner, Emilia J. Diego, Galen E. Switzer, Sara P. Myers.

**Methodology:** Katherine M. Reitz, Stephanie M. Downs-Canner, Emilia J. Diego, Sara P. Myers.

**Project administration:** Sara P. Myers.

**Resources:** Maryanna S. Owoc, Sara P. Myers.

**Software:** Clair N. Smith, Sara P. Myers.

**Supervision:** Sara P. Myers.

**Validation:** Sara P. Myers.

**Writing – original draft:** Katherine M. Reitz, Lauren Terhorst, Clair N. Smith, Insiyah K. Campwala, Maryanna S. Owoc, Stephanie M. Downs-Canner, Emilia J. Diego, Galen E. Switzer, Matthew R. Rosengart, Sara P. Myers.

**Writing – review & editing:** Katherine M. Reitz, Lauren Terhorst, Clair N. Smith, Insiyah K. Campwala, Maryanna S. Owoc, Stephanie M. Downs-Canner, Emilia J. Diego, Galen E. Switzer, Matthew R. Rosengart, Sara P. Myers.

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
