## [Decision Letter · Decision Letter 0]

13 Jul 2021

PONE-D-21-19123

Healthcare providers’ perceived support from their organization is associated with lower burnout amid the COVID-19 pandemic

PLOS ONE

Dear Dr. Myers,

Thank you for submitting your manuscript to PLOS ONE. After careful consideration, we feel that it has merit but does not fully meet PLOS ONE’s publication criteria as it currently stands. Therefore, we invite you to submit a revised version of the manuscript that addresses the points raised during the review process.

We look forward to receiving your revised manuscript.

Kind regards,

Forough Mortazavi

Academic Editor

PLOS ONE

Additional Editor Comments:

Dear authors,

Thank you for conducting this study on interesting topic. Here are the some criticisms from my evaluation:

1. The title does not cover all three significant variable.

2. The method of recruitment is not clearly described. Kindly explain why a prospective design was used for data gathering, because in research topics similar to yours, a cross-sectional study is much more common.

3. In the results section, PLS describe how the results were improved by your chosen method of data collection. Kindly explain the advantages of your method of recruitment.

4. PLS clarify if respondents received questionnaires several times.

5. Please mention the Max and Min total scores for all the scales and if there is a cut-off point for every scale. For all measures, kindly provide scale's validity and reliability indices.

6. PLS clarify the final sample size and describe how the final sample size was calculated.

7. Path analysis is usually used for examining relationships between three variables such as anxiety, POS, and burnout. Why didn’t you use path analysis?

8. PLS include the sample size for each table.

9. I noticed that some variables lack a reference value in tables 2 and 3. PLS check all variables and provide a reference value for each.

10. Kindly mention the statistical test conducted in footnote sections of the tables.

11. Please specify that in tables 2 and 3, analysis was done on which sample, 119, 402, or 516?

12. PLS mention which scales were filled out in each survey? Did each participant receive a different scale in each survey? For example, a group of participants received anxiety in survey 1, burnout in survey 2, and SOP in survey 3? If so, why did you run 6 surveys?

13. PLS mention the strong and weak points of the study as well as its novelty aspects.

Journal Requirements:

Reviewers' comments:

Reviewer's Responses to Questions

**Comments to the Author**

1. Is the manuscript technically sound, and do the data support the conclusions?

Reviewer #1: Yes

2. Has the statistical analysis been performed appropriately and rigorously? 

Reviewer #1: Yes

3. Have the authors made all data underlying the findings in their manuscript fully available?

Reviewer #1: No

4. Is the manuscript presented in an intelligible fashion and written in standard English?

Reviewer #1: Yes

5. Review Comments to the Author

Reviewer #1: I want to thank you for the opportunity to review this manuscript. The time spent creating and shipping it is greatly appreciated. In my humble opinion, it offers interesting results that the scientific community and hospital managers can benefit from. The exhaustion, disorders and psychological symptoms caused by COVID-19 and what can be done to alleviate them in the most affected group par excellence is essential. Therefore, this work, correctly planned and approached, is one more addition to this international struggle. Therefore, first of all I would like to congratulate the authors.

In the other hand, I believe that there are some aspects that can be improved. Here are my recommendations. Hopefully they are useful:

In the introduction, no mention is made of many of the sociodemographic and labor variables that were evaluated at work. This section needs an extension to theoretically justify the selection and inclusion of these variables in the analyzes. A part of this information can be transferred from the discussion section, where a multitude of references of interest are collected.

Clearly state the objectives of the study at the end of the introduction.

In the method, he points out that, according to the previous literature, it is assumed that 35% of health care providers suffered from burnout. At this point, add the corresponding reference.

Clarify further how the sample was obtained. Do the email addresses of all employees and all hospitals appear in the directory? Were there any specific criteria for the selection of hospitals?

Indicate the reliability of the instruments found for your study.

I suggest the elimination of the appendices, since all the data necessary to understand the study are correctly established in the text. However, if the authors wish to keep them, at least it would be appropriate to have the consent of the authors of the three scales used mainly in the work for their reproduction. Or, if not, delete these from the annexes.

The information on the study participants (initial number, final number, figure 1…) would be preferable to be presented in the sub-section of participants of the method. These data do not respond to the objective of the work and, therefore, do not proceed in this section. In addition, I miss that data of interest on the participantes are indicated, such as the age range or the average by gender.

Review the titles provided in table 1. Specifically, it would be recommended that n and% appear on the numerical data (not on the left). On the other hand, review the data and include decimals where appropriate (the sum of the percentages provided is greater than 100% in some cases).

In Figure 1, you speak in ordinally numbers of the surveys. While the tables talk about the surveys in months. Please choose one of the two ways to improve consistency throughout the text. In my opinion, the most appropriate thing would be to point out, for example, "First survey (April)" ...

In relation to the data on gender, a possible variable implicated in the disparity of results is the job position. Since this is a study of healthcare providers with a wide variety of occupations, it would be advisable to mention this potential confounder or add it as a limitation.

In the conclusions of the work, add possible practical applications of the findings.

Thanks for your work.

6. PLOS authors have the option to publish the peer review history of their article (what does this mean?). If published, this will include your full peer review and any attached files.

Reviewer #1: No

---

## [Author Response · Author response to Decision Letter 0]

20 Aug 2021

Response to reviewers:

PLOS One Additional Editor Comments:

1. The title does not cover all three significant variable.

Response: We thank the editor for his request that the title accurately reflect the objectives and findings of our study. The title now includes reference to all three variables considered.

Related revised text: Title: “Healthcare providers’ perceived support from their organization is associated with lower burnout and anxiety amid the COVID-19 pandemic”

2. The method of recruitment is not clearly described. Kindly explain why a prospective design was used for data gathering, because in research topics similar to yours, a cross-sectional study is much more common.

Response: We welcome the opportunity to elaborate on our a priori study design and method of recruitment and feel strongly that the prospective longitudinal nature of this study offers several advantages. First, risk of burnout evolves over time; career deprioritization and emotional exhaustion are responses to repetitive insults (Bakker AB, Costa PL. Chronic job burnout and daily functioning: A theoretical analysis. Burnout Research. 2014; 1(3):112-119). The protracted nature of this pandemic has been shown to have negative long-term psychosocial consequences, which can best be understood prospectively by a longitudinal study design (McBride O, Butter S, Murphy J, et al. Context, design and conduct of the longitudinal COVID-19 psychological research consortium study-wave 3. Psychiatric Research. 2021; e1880: 1-17. Second, a longitudinal design in this setting allows for understanding how individuals may react to changes in organizational dynamics, restrictions, and protocols that occurred in response to the COVID-19 crisis, especially during the early period of the pandemic when uncertainty about the disease, contagion, and healthcare resources was greatest. Third, we were especially interested in understanding how all levels of healthcare providers were influenced by these issues. Unlike existing studies, which have made important contributions to our knowledge of mental health during COVID-19 by focusing on specific types of providers (e.g., professional staff nurses, physicians, etc.) or all providers within a system without comparing occupation-types to each other, our study now sheds light on how frontline healthcare workers with different levels of patient-care responsibilities may react differently during times of organizational and global health crisis. This nuanced perspective will aid in our ability to improve healthcare team dynamics and optimize patient care. Despite these strengths, we acknowledge that the longitudinal study design contributed to participant attrition, which has consequences for study validity and the representativeness of our sample. We have revised the text, as indicated below, to discuss these issues and provide a more transparent rationale for our recruitment strategy. 

Related revised text: Methods, Participants, Settings, Study Design subsection, page 6, paragraph 1: In order to assess how pandemic-related changes in organizational dynamics and protocols may influence anxiety and risk for burnout over time, an online survey of previously validated measures was administered monthly for a six-month period (April 28 to October 11, 2020).

Page 6-7, paragraph 2: In order to better understand how healthcare workers with differing levels of patient-care responsibilities may have reacted to this organizational and global health crisis, we sampled a variety of providers; attending physicians, physicians in training (i.e., residents, fellows), advanced practice providers (i.e., nurse practitioners, physician assistants), nurses, and other providers (i.e., respiratory therapists, patient care technicians) employed by an UPMC-associated hospital in Pennsylvania with sufficient exposure to work-place and institutional culture (i.e., >12 months of employment) were eligible for inclusion.

3. In the results section, PLS describe how the results were improved by your chosen method of data collection. Kindly explain the advantages of your method of recruitment.

Response: The authors thank this editor for his suggestion to describe the advantages of a longitudinal study design and our chosen method of data collection. We do, however, respectfully feel that this description is better suited for the discussion section, and, specifically, the subsection in which the study’s limitations and strengths are outlined. As this Editor has requested a more robust examination of the study’s strengths and weaknesses elsewhere (Comment# 13), we have expanded this subsection.

Related revised text: Discussion, page 18, paragraph 3: Although several provider-types were recruited in an attempt to understand how individuals with different patient-care responsibilities may have been variably influenced by pandemic-related restrictions and we surveyed individuals from multiple hospitals, our data does represent a single healthcare system and geographical area, which may limit external validity. While conducting a prospective longitudinal study allowed for the ability to assess providers’ responses to changes in organizational support, anxiety, and risk for burnout over time, this methodology also likely contributed to high attrition rates at later survey administrations, presumably due to survey fatigue. 

4. PLS clarify if respondents received questionnaires several times.

Response: Participants were initially randomized to receiving one of two versions of the survey. Randomization was stratified based on gender, whether the participant was employed by a UPMC-affiliated academic vs. community practice, and occupation type. These stratification criteria were decided on a priori as factors that may influence anxiety/burnout as well as degree or organizational support. The rationale for randomization was to minimize response bias due to context effects, which may occur if an individual participant’s responses to survey items are influenced by the order in which they appear within the survey instrument (please see Methods, page 7 and associated reference). The exposure of interest, perceived organizational support, was the initial subscale in both survey versions. Survey versions A and B differed insofar as one presented the anxiety assessment scale second and the burnout assessment third while the alternate version had the reverse order of these two subscales. Randomization occurred at the beginning of the study, after which the participant received the same version of the survey for all six survey administrations. This methodology and study design was undertaken after consultation with our co-author, Dr. Galen Switzer, an expert in qualitative and psychometric research. 

Related revised text: Methods, page 7, paragraph 1: Participants received the same version at all six time points via an automated email providing a link to a University of Pittsburgh RedCap database where de-identified data was collected, stored, and connected to the individuals’ email for repeated measure.

5. Please mention the Max and Min total scores for all the scales and if there is a cut-off point for every scale. For all measures, kindly provide scale's validity and reliability indices.

Response: The authors appreciate this request for clarification. The manuscript has been revised to include the maximum and minimum total scores for each scale (please see Methods, Study Measures). It is the authors’ practice to report the internal consistency of validated instruments (i.e., reliability) and to reference original manuscripts describing validation of the survey measures used for further details. We have referenced the reliability of the original 8-item Survey of Perceived Organizational Support, the 10-item Burnout scale from the Professional Quality (Pro-QOL) scale, and the 4-item Emotional Distress-Anxiety short form of the Patient Reported Outcome Measurement Information System (PROMIS) scale as the internal consistency, i.e., Cronbach’s alpha, for each of these instruments. This was stated in the original text in paragraph 1, page 7: “Internal consistency of each scale or subscale are as follows: 8-item SPOS=0.93,14 Burnout=0.84,15 and Anxiety= 0.97.16” The authors would be happy to include additional details about validation of the original survey instruments, but respectfully feel that this is out of the scope of our current manuscript.

Related revised text: Methods, Page 7, paragraph 2: POS was assessed using the validated, unidimensional 8-item Survey of POS (SPOS) scale.9,10 Items appraise the degree to which an individual agreed with statements describing support or commitment from their employer organization in the 30 days prior using a 7-point Likert scale (0=Strongly disagree, 6=Strongly agree; scale minimum=0, maximum=42).

Methods, Page 7, paragraph 2 through page 8, paragraph 1: Risk for burnout was evaluated using the 10-item Burnout scale from the Professional Quality (Pro-QOL) scale. This instrument is a validated, multidimensional metric that assesses consequences of stress a provider may experience from exposure to patients in emotional or physical distress.11,12 Participants reflect on the frequency with which they experience hopelessness or difficulty completing job-related tasks effectively in the prior 30-days using a 5-point Likert scale (1=Never, 5=Very often; scale minimum=5, maximum=30). General anxiety was assessed using the 4-item Emotional Distress-Anxiety short form of the validated, multidimensional Patient Reported Outcome Measurement Information System (PROMIS)13 scale that evaluates the frequency to which anxiety-related symptoms were experienced in the prior 7-days using a 5-point Likert scale (1=Never, 5=Very often; scale minimum=7, maximum=35).

6. PLS clarify the final sample size and describe how the final sample size was calculated.

Response: Thank you for the opportunity to clarify our data. We have updated the content within Figure 1 to improve the clarity of our primary cohort and those included in the sensitivity analysis. 

Related revised text:

Figure 1. Cohort accrual

1 Discrepancy between number excluded from primary analysis and number lost to follow-up stems from participants failing to complete consecutive surveys. As data from participants who completed two of the three initial surveys was included in the primary analysis, some individuals may have cursorily been considered lost to follow up after survey 1 but been included in the analysis if they returned to complete survey 3. 

2 Within survey missingness.

7. Path analysis is usually used for examining relationships between three variables such as anxiety, POS, and burnout. Why didn’t you use path analysis?

Response: A mediation analysis was used to demonstrate if anxiety was the mechanism through which perceived organizational support transmits its effect on burnout. Although we agree that path analysis is one way to test mediation, along with regression and structural equation models, our decision to use regression was based on our hypothesis that anxiety mediated the observed relationship between POS and burnout. If we were to expand the model to examine other potential mediators, a path analysis or structural equation model would be the more appropriate analysis. We do, however, respectfully maintain that the chosen analysis is fitting given our hypothesis.

Related revised text: Not applicable.

8. PLS include the sample size for each table.

Response: Each table has been updated in accordance with this suggestion.

Related revised text: Tables 1-3.

9. I noticed that some variables lack a reference value in tables 2 and 3. PLS check all variables and provide a reference value for each.

Response: Each table has been updated in accordance with this suggestion.

Related revised text: Tables 1-3. 

10. Kindly mention the statistical test conducted in footnote sections of the tables.

Response: Each table has been updated in accordance with this suggestion.

Related revised text: Tables 1-3. 

11. Please specify that in tables 2 and 3, analysis was done on which sample, 119, 402, or 516?

Response: Each table has been updated in accordance with this suggestion.

Related revised text: Tables 2, 3.

12. PLS mention which scales were filled out in each survey? Did each participant receive a different scale in each survey? For example, a group of participants received anxiety in survey 1, burnout in survey 2, and SOP in survey 3? If so, why did you run 6 surveys?

Response: Each survey administration included all three subscales: perceived organizational support, burnout, and anxiety. Participants all received the same subscales within the version of the survey that they were randomized to receive. Please see response to Editor’s comment#4. 

Related revised text: Not applicable. 

13. PLS mention the strong and weak points of the study as well as its novelty aspects.

Response: We have revised the text of the manuscript based on this and other recommendations based on the editor’s and reviewer’s feedback. While there are several investigations examining the psychological effects of the COVID-19 pandemic on healthcare providers, to our knowledge these studies rarely include analysis by provider subtype. We believe that this distinction is important and contributes to the novelty of this study because differences in type of patient-care responsibilities may impact the psychological ramifications of providing pandemic-related care. Additionally, the prospective and longitudinal nature of this study is unique. Unlike most survey-based research, which as this editor points out is cross-sectional, our prospective and longitudinal inquiry allows a more accurate understanding of the protracted nature of this pandemic may have influenced perceived organizational support, anxiety, and burnout over time. Additionally, as we recognize that responses to survey items may be influenced by preceding items, thereby serving as a source of bias, participants were randomized to receiving one of two versions of our survey after being stratified based on provider type, sex, and affiliated with an academic versus community hospital. As discussed elsewhere, these versions differed only in the order in which survey subscales appeared. Participants received the same designated version of the survey for each of the six survey administrations. Unfortunately, despite our best efforts, we acknowledge limitations of having surveyed providers within a single institution and the attrition and missingness of data that are inherent to longitudinal survey studies. 

Related revised text: Discussion, page 18 paragraph 3- page 19 paragraph 1. There are several strengths and limitations worth addressing. Although several provider-types were recruited in an attempt to understand how individuals with different patient-care responsibilities may have been variably influenced by pandemic-related restrictions and we surveyed individuals from multiple hospitals, our data does represent a single healthcare system and geographical area, which may limit external validity. Efforts were made to reduce response bias by randomizing individuals to receiving one of two versions of our survey, which differed in the order of the constructs assessed. While conducting a prospective longitudinal study allowed for the ability to assess providers’ responses to changes in organizational support, anxiety, and risk for burnout over time, this methodology also likely contributed to high attrition rates at later survey administrations, presumably due to survey fatigue. Therefore, missingness may not be at random, forcing respondent exclusion from the primary analysis. Further, the racial/ethnic homogeneity of our sample likely compromised our ability to accurately assess how sociodemographic factors influence burnout and anxiety in the context of a multifaceted construct such as POS. As this study was executed during the early months of the pandemic, variability in diagnostic testing across the healthcare system may have contributed to underreporting of hospital-specific rates of COVID-19. Finally, survey response was adequately powered to evaluate the a priori hypotheses; however, exploratory moderation analyses were underpowered.

 

Reviewer comments:

Reviewer #1 (Remarks to the author): I want to thank you for the opportunity to review this manuscript. The time spent creating and shipping it is greatly appreciated. In my humble opinion, it offers interesting results that the scientific community and hospital managers can benefit from. The exhaustion, disorders and psychological symptoms caused by COVID-19 and what can be done to alleviate them in the most affected group par excellence is essential. Therefore, this work, correctly planned and approached, is one more addition to this international struggle. Therefore, first of all I would like to congratulate the authors. In the other hand, I believe that there are some aspects that can be improved. Here are my recommendations. Hopefully they are useful.

The authors very much appreciate this positive feedback. We have done our best to address the critiques of this reviewer and the comments provided by the Editor. It is our belief that the revised manuscript is markedly improved.

1. In the introduction, no mention is made of many of the sociodemographic and labor variables that were evaluated at work. This section needs an extension to theoretically justify the selection and inclusion of these variables in the analyzes. A part of this information can be transferred from the discussion section, where a multitude of references of interest are collected.

Response: The authors understand the need to clarify and justify the sociodemographic variables included in the study and analysis. These variables included age, gender, marital status, race/ethnicity, occupation, hospital type (academic vs. community), income, and dependent status. We now briefly state the importance of these variables in influencing the outcomes of interest (anxiety and burnout) and call attention to relevant references. In an effort to maintain a succinct and clear introduction, however, these variables and the key findings of this study are explored more fully in the discussion. 

Related revised text: Introduction, page 5, paragraph 1: While burnout has been described as a reaction to organizational factors such as work-related stress,3 individual factors including sociodemographic variables associated with development of coping skills in response to stressful situations (e.g., age), availability of support systems (marital status), and type and degree of responsibility (e.g., occupation, parental/caretaker status, etc.) may increase susceptibility to burnout4 and/or anxiety.5

References associated with the above revised text are: 

Green AE, Albanese BJ, Shaprio NM, Aarons GA. The roles of individual and organizational factors in burnout among community-based mental health service providers. Psychol Serv. 2014; 11(1):41-49.

Hubbard G, den Daas C, Johnston M, Dixon D. Sociodemographic and psychological risk factors for anxiety and depression: Findings from the COVID-19 Health and Adherence research in Scotland on Mental Health (CHARIS-MH) cross-sectional survey. International Journal of Behavioral Medicine. 2021; open access.

2. Clearly state the objectives of the study at the end of the introduction.

Response: The authors have revised the introduction to plainly state the aims of this study.

Related revised text: Introduction, page 5, paragraph 1: In this longitudinal prospective study of healthcare providers, we aim to evaluate whether 1) POS is associated with anxiety or burnout, hypothesizing that during the COVID-19 pandemic higher levels of POS are associated with lower burnout and anxiety, and 2) anxiety mediates the association between POS and burnout.

3. In the method, he points out that, according to the previous literature, it is assumed that 35% of health care providers suffered from burnout. At this point, add the corresponding reference.

Response: The text has been revised so that the references for the prevalence of burnout among healthcare providers is included. The target sample size calculation/formula was based on a recent study about mental health outcomes among frontline healthcare providers exposed to COVID-19. This reference, which was originally cited at the end of the sentence, is retained in the text. 

Related revised text: Methods, page 6, paragraph 2: Reference added- Lee AM, Wong JG, McAlonan GM, et al. Stress and psychological distress among SARS survivors 1 year after the outbreak. Can J Psychiatry. 2007;52(4):233-240.

4. Clarify further how the sample was obtained. Do the email addresses of all employees and all hospitals appear in the directory? Were there any specific criteria for the selection of hospitals?

Response: The authors appreciate the request for a more detailed description of how participants were recruited. A request for participation (Appendix A) was e-mailed to all full-time providers employed by the UPMC healthcare system. E-mail addresses were obtained from an internal employee electronic directory that is available to and may be accessed by any UPMC staff. This directory is maintained by the institution’s Office of Human Resources such that the email addresses of all providers at all hospitals are included in the directory. Three study team members (SM, IS, and MO) reviewed the entire directory to abstract all providers who were listed as either attending physicians, residents/fellows, advanced practice providers, professional staff nurses, respiratory therapists, or patient care technicians. Not including expired/nonworking email addresses, 9,578 providers were emailed. As mentioned in the methods section, UPMC is a large, multihospital healthcare system that regularly acquires additional hospitals. All hospitals in Pennsylvania recognized as UPMC facilities as of January 1, 2020 were included in this study. 

Related revised text: Methods, page 6, paragraph 2: All full-time providers employed by the healthcare system were recruited for study participation via e-mail (Appendix A). E-mail addresses were abstracted by three study investigators (IS, MO, and SPM) from an internal employee electronic directory (April 1-14, 2020) which is available to all UPMC staff and is maintained by the institution’s Office of Human Resources. Attending physicians, physicians in training (i.e., residents, fellows), advanced practice providers (i.e., nurse practitioners, physician assistants), nurses, and other providers (i.e., respiratory therapists, patient care technicians) employed by an UPMC-associated hospital in Pennsylvania with sufficient exposure to work-place and institutional culture (i.e., >12 months of employment) were eligible for inclusion.

5. Indicate the reliability of the instruments found for your study.

Response: While we recognize that there are various methods of assessing both reliability and validity of existing survey instruments, we have referenced the reliability of the original 8-item Survey of Perceived Organizational Support, the 10-item Burnout scale from the Professional Quality (Pro-QOL) scale, and the 4-item Emotional Distress-Anxiety short form of the Patient Reported Outcome Measurement Information System (PROMIS) scale as the internal consistency, i.e., Cronbach’s alpha, for each of these instruments. This was stated in the original text in paragraph 1, page 7: “Internal consistency of each scale or subscale are as follows: 8-item SPOS=0.93,14 Burnout=0.84,15 and Anxiety= 0.97.16” The authors would be happy to include additional details about validation of the original survey instruments, but respectfully feel that this is out of the scope of our current manuscript. 

Related revised text: Not applicable. 

6. I suggest the elimination of the appendices, since all the data necessary to understand the study are correctly established in the text. However, if the authors wish to keep them, at least it would be appropriate to have the consent of the authors of the three scales used mainly in the work for their reproduction. Or, if not, delete these from the annexes.

Response: While the authors have strived to include all the materials/data germane to comprehending the study and its analysis within the text of the manuscript, as this reviewer graciously acknowledges, we do feel there is benefit in including additional information in the appendices for the sake of completeness and transparency. Nevertheless, we fully appreciate the importance of requesting permission for the publication of the original three scales from the instruments’ authors. For this reason, as this reviewer has suggested, we have eliminated the text of these scales from the appendices. 

Related revised text: Supplemental material.

7. The information on the study participants (initial number, final number, figure 1…) would be preferable to be presented in the sub-section of participants of the method. These data do not respond to the objective of the work and, therefore, do not proceed in this section. In addition, I miss that data of interest on the participantes are indicated, such as the age range or the average by gender.

Response: The authors thank this reviewer for the suggestion to improve the intelligibility of our manuscript. We feel, however, that this may be a stylistic issue and would like to, with the permission of both the reviewer and the editor, maintain Figure 1’s current location in the manuscript. After consideration, although we agree with this reviewer that the content of the figure is not necessarily related directly to the execution of study objectives and results, it does provide a description of cohort accrual and the participants included in primary/sensitivity analyses. In this sense, we feel that the information is more relevant to the results from the analysis rather than the methods and study design, which were conceived of a priori and before the study commenced.

To address this reviewer’s second request, to present the demographic data of interest for all the participants rather than for just those included in the primary analysis, we wish to call attention to eTable2 in the supplement. This table describes the demographic variables for participants included and excluded from the primary analysis and also compares whether there is a statistically significant difference between these participants on the basis of any given variable. Although we considered inserting this table into the manuscript text rather than including it in the supplemental material, we feel it would detract from the primary objectives and findings. For this reason, we have chosen to retain the supplemental material (please see our response to Reviewer comment #6 above) and include the requested information within eTable 2. 

Related revised text: not applicable. Requested data, which was present in original manuscript and supplemental text, can be found in Figures 1 and eTable2.

8. Review the titles provided in table 1. Specifically, it would be recommended that n and% appear on the numerical data (not on the left). On the other hand, review the data and include decimals where appropriate (the sum of the percentages provided is greater than 100% in some cases).

Response: We have revised the table according to this reviewer’s comment. Both the titles and the sum of percentages have been edited. 

Related revised text: Table 1

9. In Figure 1, you speak in ordinally numbers of the surveys. While the tables talk about the surveys in months. Please choose one of the two ways to improve consistency throughout the text. In my opinion, the most appropriate thing would be to point out, for example, "First survey (April)" ...

Response: The authors appreciate this suggestion and have revised both the Figures and Tables so that the survey iteration is indicated both by number and month administered (e.g., 1st Survey, April 2020).

Related revised text: Figure 1, eTable5, eTable6, eTable7.

10. In relation to the data on gender, a possible variable implicated in the disparity of results is the job position. Since this is a study of healthcare providers with a wide variety of occupations, it would be advisable to mention this potential confounder or add it as a limitation.

Response: The authors appreciate this nuanced reflection on our data and agree that disparate gender composition within each provider-type category may influence our conclusions about how men and women differ with respect to anxiety and risk for burnout. This may be especially true given that structural bias related to gender may impact the degree to which women and men feel they have control over workplace processes. As this reviewer recommends, the manuscript has been revised to include this in the discussion.

Related revised text: Discussion, page 17, paragraph 2. These conflicting findings may indicate that confounders obfuscate whether gender is independently associated with burnout. One such issue, which is pertinent to our analysis, is the disparate gender composition among different provider-types. Although studies are inconsistent in their findings, lack of control over workplace practices, which is necessarily influenced by gender-based structural bias, may contributed to differences in anxiety and risk for burnout in specialties where women are underrepresented.

The following reference has been added in association with the above revised text: Purvanova RK, Muros JP. Gender differences in burnout: A meta-analysis. Journal of Vocational behavior. October 2010. 77(2):168-185. DOI:10/1016/j.jvb.2010.04.006.

11. In the conclusions of the work, add possible practical applications of the findings.

Response: The authors agree that explicitly stating the practical applications of our findings will strengthen the manuscript. The association between perceived organizational support and anxiety and risk for burnout indicate that feeling supported by one’s employer can provide comfort during stressful circumstances, e.g., the COVID-19 pandemic. Implementing discretionary supportive services such as benefits and personalizing these services based on individual factors, some of which are highlighted in this manuscript, may increase the degree to which employees feel supported. Other methods may include those that improve communication between supervisors and their employees, especially with respect to transparency in the adoption and oversight of institutional policies in response to crisis. 

Related revised text: Conclusion, page 19, paragraph 1. These data indicate that interventions aimed at increasing perceived organizational support, such as augmenting discretionary services and benefits (e.g., childcare, psychological-support services, etc.), or fostering transparency in the adoption and oversight of institutional policies in response to crisis, may promote provider well-being.

---

## [Decision Letter · Decision Letter 1]

27 Sep 2021

PONE-D-21-19123R1Healthcare providers’ perceived support from their organization is associated with lower burnout and anxiety amid the COVID-19 pandemicPLOS ONE

Dear Dr. Myers,

Thank you for submitting your manuscript to PLOS ONE. After careful consideration, we feel that it has merit but does not fully meet PLOS ONE’s publication criteria as it currently stands. Therefore, we invite you to submit a revised version of the manuscript that addresses the points raised during the review process.

We look forward to receiving your revised manuscript.

Kind regards,

Forough Mortazavi

Academic Editor

PLOS ONE

Journal Requirements:

Additional Editor Comments (if provided):

Dear authors,

Thank you for revising the manuscript; however, a few points raised in my evaluation still remain:

1. Ethical considerations are missing from the methods section. PLS kindly add them.

2. PLS clarify if respondents received anonymous questionnaires.

3. Page 16, the highlighted lines are suitable for the introduction section. 4. The manuscript must be edited so that it follows scientific writing standards.

Reviewers' comments:

Reviewer's Responses to Questions

**Comments to the Author**

1. If the authors have adequately addressed your comments raised in a previous round of review and you feel that this manuscript is now acceptable for publication, you may indicate that here to bypass the “Comments to the Author” section, enter your conflict of interest statement in the “Confidential to Editor” section, and submit your "Accept" recommendation.

Reviewer #1: All comments have been addressed

2. Is the manuscript technically sound, and do the data support the conclusions?

Reviewer #1: Partly

3. Has the statistical analysis been performed appropriately and rigorously? 

Reviewer #1: Yes

4. Have the authors made all data underlying the findings in their manuscript fully available?

Reviewer #1: Yes

5. Is the manuscript presented in an intelligible fashion and written in standard English?

Reviewer #1: Yes

6. Review Comments to the Author

Reviewer #1: Dear authors,

I appreciate the work to attend to the previous suggestions. Although most of the questions have been approached or answered appropriately, I consider that some of the points made in the previous review have not been addressed or, at least, not in depth. For this reason, I reiterate two of the recommendations made previously:

-The introduction needs a theoretical justification of the study. Place special emphasis on clarifying why the analyzed variables are chosen, establishing where the work comes from and why it is important to address this. It may not have been analyzed how employer support affects burnout in workers during COVID-19, but what about other health crises or situations of high workload? Thus, I insist on the need to substantially improve the introduction. The discussion of the findings should be supported by a good initial introduction.

-Indicate the reliability of the instruments found in your study. It is understood that the reliability of the original scales is adequate, since otherwise they would not be valid instruments. However, what should be noted in the manuscript is the level of reliability of the scales for your sample.

Thank you for your work.

7. PLOS authors have the option to publish the peer review history of their article (what does this mean?). If published, this will include your full peer review and any attached files.

Reviewer #1: No

---

## [Author Response · Author response to Decision Letter 1]

11 Oct 2021

PLOS One Additional Editor Comments:

1. Ethical considerations are missing from the methods section. PLS kindly add them.

Response: The authors appreciate the request for transparency with regard to ethical considerations in study design and execution. We have revised the text of the methods section to specifically address the following: confidentiality and anonymity, informed consent, adverse events and benefits, and ability to withdraw from the study. Ethical considerations that were previously mentioned, including IRB approval and incentives, have been retained. The text of the recruitment script and informed consent document are present in the supplemental material. We feel that the manuscript now completely addresses ethical considerations.

Related revised text: 

Methods: page 8, paragraph 1: “Participants were provided with study details, purpose of the investigation, and incentive information in the text of the recruitment e-mail and within the consent document. Participants were made aware that although the investigators did not anticipate that the study would cause adverse events, responding to survey items may evoke emotional responses such as stress. There were no direct benefits to the participant for completing surveys. Participants were free to withdraw from the study at any point; in this circumstance, all contact and data collection would cease and any existing data would be deleted upon request.” 

2. PLS clarify if respondents received anonymous questionnaires.

Response: As requested above, we have clarified anonymization of the data. The participants were originally contacted via their email address. These email addresses were used to assign participants an alphanumeric code so that their serial data could be collected and stored in an anonymous fashion. One of the study investigators (IC) had access to the document that linked the email addresses to the anonymized participant code. To blind her to participant data, she did not have access to the stored data and was not involved with the statistical analysis. 

Related revised text: Please see response to Editor’s comment # 1 above.

3. Page 16, the highlighted lines are suitable for the introduction section. 

Response: We thank the editor for this suggestion. The authors believe that revising the manuscript to include the highlighted lines in the introduction rather than the discussion addresses comment #1 of the reviewer. Specifically, the introduction now clearly describes the importance of studying burnout and anxiety. We relate the significance of burnout to reduced productivity, adverse clinical outcomes, increased rate of medical errors, and the financial strain on the US economy. As per the request of the reviewer, we describe how existing literature has described the role of heavy workloads, changing clinical rolls, etc. on stress and burnout. The role of anxiety on burnout and during periods of crisis is also introduced. Finally, we have contextualized the influence of perceived organizational support on these constructs during the COVID-19 pandemic. 

Related revised text: Introduction, Page 5, Paragraphs 2. “Burnout among healthcare providers is a prevalent and well-documented phenomenon3 and has been associated with adverse clinical outcomes,4 reduced productivity,5 and increased rate of medical errors.6 Aside from the consequences to providers and their patients, burnout represents a serious financial burden; $4.6 billion lost annually is attributed to physician attrition and loss of clinical hours secondary to burnout.6 Heavy workloads, changing clinical roles, reduced decision latitude (i.e., the ability to exercise control over work-related responsibilities), and lack of support from supervisors have been cited as causes of increased occupational stress,7 a key predictor of burnout.8 During this pandemic additional stressors such as lack of personal protective equipment and fear of contagion have been shown to augment the risk for burnout.1,3 These issues have threatened the stability of our healthcare organizations and, therefore, the COVID-19 medical crisis has also functionally become an organizational crisis.9 As a result, health-related performance and quality outcomes may, in part, depend on how a healthcare organization manages to allocate resources and human capital10 and whether it effectively develops policies to protect and support its employees.”

4. The manuscript must be edited so that it follows scientific writing standards.

Response: We have reviewed the submission guidelines on the PLOS ONE webpage. We have added line numbers to the manuscript file using continuous numbering. We have ensured that all abbreviations were defined upon first appearance in the text and that non-standard abbreviations were used only when they appeared three or more times in the test. We have consulted the Vancouver style guidelines for citing references to ensure that the format is correct. We have removed the reference to the following article, which has been retracted: Panagioti M, Geraghty K, Johnson J, et al. Association between physician burnout and patient safety, professionalism, and patient satisfaction: a systematic- review and meta-analysis. JAMA Intern Med. 2018; 178:1317-1330. doi: 10.1001/jamainternmed.2018.3713. The manuscript organization has been revised so that the acknowledgements and references appear in the acceptable order at the end of the manuscript. The title page has been revised according to the sample provided on the PLOS ONE webpage. Titles/professional credentials have been removed. 

Related revised text: Please see title page, reference list, and ending comments. 

 

Reviewer comments:

Reviewer #1 (Remarks to the author): I appreciate the work to attend to the previous suggestions. Although most of the questions have been approached or answered appropriately, I consider that some of the points made in the previous review have not been addressed or, at least, not in depth. For this reason, I reiterate two of the recommendations made previously:

1. The introduction needs a theoretical justification of the study. Place special emphasis on clarifying why the analyzed variables are chosen, establishing where the work comes from and why it is important to address this. It may not have been analyzed how employer support affects burnout in workers during COVID-19, but what about other health crises or situations of high workload? Thus, I insist on the need to substantially improve the introduction. The discussion of the findings should be supported by a good initial introduction.

Response: We apologize for not sufficiently addressing this in the initial revisions. The authors agree with the suggestion from this reviewer. We have revised the manuscript text to incorporate these details. Specifically, the introduction now clearly describes the importance of studying burnout and anxiety. We relate the significance of burnout to reduced productivity, adverse clinical outcomes, increased rate of medical errors, and the financial strain on the US economy. As per the request of the reviewer, we describe how existing literature has described the role of heavy workloads, changing clinical rolls, etc. on stress and burnout. The role of anxiety on burnout and during periods of crisis is also introduced. Finally, we have contextualized the influence of perceived organizational support on these constructs during the COVID-19 pandemic.

 Related revised text: Introduction, Page 5, Paragraphs 1-2.

2. Indicate the reliability of the instruments found in your study. It is understood that the reliability of the original scales is adequate, since otherwise they would not be valid instruments. However, what should be noted in the manuscript is the level of reliability of the scales for your sample.

Response: We thank this reviewer for their comment. The internal consistency of the subscales using participants’ data have been calculated and added to the results section.

Related revised text: Results, Page 11, Paragraph 1: “Internal consistency calculated for subscales was as follows: SPOS=0.91, Burnout=0.85, and Anxiety= 0.89.”

---

## [Decision Letter · Decision Letter 2]

28 Oct 2021

Healthcare providers’ perceived support from their organization is associated with lower burnout and anxiety amid the COVID-19 pandemic

PONE-D-21-19123R2

Dear Dr. Myers,

We’re pleased to inform you that your manuscript has been judged scientifically suitable for publication and will be formally accepted for publication once it meets all outstanding technical requirements.

Kind regards,

Forough Mortazavi

Academic Editor

PLOS ONE

Additional Editor Comments (optional):

Reviewers' comments:

Reviewer's Responses to Questions

**Comments to the Author**

1. If the authors have adequately addressed your comments raised in a previous round of review and you feel that this manuscript is now acceptable for publication, you may indicate that here to bypass the “Comments to the Author” section, enter your conflict of interest statement in the “Confidential to Editor” section, and submit your "Accept" recommendation.

Reviewer #1: All comments have been addressed

2. Is the manuscript technically sound, and do the data support the conclusions?

Reviewer #1: Yes

3. Has the statistical analysis been performed appropriately and rigorously? 

Reviewer #1: Yes

4. Have the authors made all data underlying the findings in their manuscript fully available?

Reviewer #1: Yes

5. Is the manuscript presented in an intelligible fashion and written in standard English?

Reviewer #1: Yes

6. Review Comments to the Author

Reviewer #1: I thank the authors for attend the recommendations. I believe that the manuscript has improved its quality and can be published. However, if possible, I would recommend that the authors include the reliability of the instruments in the instruments section (not in the results).

Beyond this recommendation, I can only congratulate the authors. Good job!

7. PLOS authors have the option to publish the peer review history of their article (what does this mean?). If published, this will include your full peer review and any attached files.

Reviewer #1: No

---

## [Editor Report · Acceptance letter]

11 Nov 2021

PONE-D-21-19123R2 

Healthcare providers’ perceived support from their organization is associated with lower burnout and anxiety amid the COVID-19 pandemic 

Dear Dr. Myers:

I'm pleased to inform you that your manuscript has been deemed suitable for publication in PLOS ONE. Congratulations! Your manuscript is now with our production department. 

Kind regards, 

on behalf of

Dr. Forough Mortazavi 

Academic Editor

PLOS ONE